# Dynamic regulation of transcription factors by nucleosome remodeling

Ming Li[1], Arjan Hada[2], Payel Sen[2], Lola Olufemi[2], Michael A Hall[3†], Benjamin Y Smith[3‡], Scott Forth[3§], Jeffrey N McKnight[4¶], Ashok Patel[4#], Gregory D Bowman[4], Blaine Bartholomew[2], Michelle D Wang[3*]

[1]Department of Chemistry and Chemical Biology, Cornell University, Ithaca, United States; [2]Department of Biochemistry and Molecular Biology, Southern Illinois University School of Medicine, Carbondale, United States; [3]Department of Physics, Laboratory of Atomic and Solid State Physics, Howard Hughes Medical Institute, Cornell University, Ithaca, United States; [4]Thomas C. Jenkins Department of Biophysics, Johns Hopkins University, Baltimore, United States

*For correspondence: mwang@ physics.cornell.edu

Present address: †Microsoft Incorporation, Redmond, United States; ‡Google Inc, New York, United States; §Laboratory of Chemistry and Cell Biology, Rockefeller University, New York, United States; ¶Fred Hutchinson Cancer Research Center, Seattle, United States; #Kusuma School of Biological Sciences, Indian Institute of Technology Delhi, New Delhi, India

**Competing interests:** The authors declare that no competing interests exist.

**Abstract** The chromatin landscape and promoter architecture are dominated by the interplay of nucleosome and transcription factor (TF) binding to crucial DNA sequence elements. However, it remains unclear whether nucleosomes mobilized by chromatin remodelers can influence TFs that are already present on the DNA template. In this study, we investigated the interplay between nucleosome remodeling, by either yeast ISW1a or SWI/SNF, and a bound TF. We found that a TF serves as a major barrier to ISW1a remodeling, and acts as a boundary for nucleosome repositioning. In contrast, SWI/SNF was able to slide a nucleosome past a TF, with concurrent eviction of the TF from the DNA, and the TF did not significantly impact the nucleosome positioning. Our results provide direct evidence for a novel mechanism for both nucleosome positioning regulation by bound TFs and TF regulation via dynamic repositioning of nucleosomes.

## Introduction

Dynamic access to specific genetic information is critical for cellular development and response to the environment. Thus, processes such as transcription must be mediated by mechanisms that regulate gene function rapidly and reliably (*Barrera and Ren, 2006*; *Kornberg, 2007*). In eukaryotic cells, proper transcriptional regulation depends upon transcription factors (TFs) that bind to specific DNA-binding sites (*Kadonaga, 2004*). Additionally, the repression of transcription has often been correlated with the presence of nucleosomes, the basic units of chromatin structure, in which histone–DNA interactions act as a barrier for RNA polymerase elongation along DNA (*Li et al., 2007*; *Petesch and Lis, 2012*; *Teves et al., 2014*). Therefore, understanding the relationship between TF binding and nucleosomes is essential in understanding gene expression and regulation (*Voss and Hager, 2014*).

Chromatin landscape and promoter architecture are dominated by the interplay of nucleosome and TF binding. Nucleosomes and TFs have been shown to compete for binding to DNA (*Mirny, 2010*; *Moyle-Heyrman et al., 2011*; *Lickwar et al., 2012*). This competition is based on the respective affinities of the TF and nucleosome for DNA, and depends upon DNA sequence, histone variants, and histone modifications. However, a nucleosome may also be repositioned through the action of chromatin remodelers, suggesting additional levels of transcription regulation. Some TFs are known to recruit nucleosome remodelers. Previous earlier studies focused on how these recruiting TFs affect the outcomes of nucleosome remodeling (*Nagaich et al., 2004*; *Boeger et al., 2008*; *Dechassa et al., 2010*; *Voss et al., 2011*). However, it is unclear how TFs that do not recruit remodelers

**eLife digest** Cells contain thousands of genes that are encoded by molecules of DNA. In yeast and other eukaryotic organisms, this DNA is wrapped around proteins called histones to make structures called nucleosomes. This compacts the DNA and allows it to fit inside the tiny nucleus within the cell. The positioning of the nucleosomes influences how tightly packed the DNA is, which in turn influences the activity of genes. Less active genes tend to be found within regions of DNA that are tightly packed, while more active genes are found in less tightly packed regions.

To activate a gene, proteins called transcription factors bind to a section of DNA within the gene called the promoter. Enzymes known as 'chromatin remodelers' can alter the locations of nucleosomes on DNA to allow the transcription factors access to the promoters of particular genes. In yeast, the SWI/SNF family of chromatin remodelers can disassemble nucleosomes to promote gene activity, while the ISW1 family organises nucleosomes into closely spaced groups to repress gene activity. However, it is not clear if, or how, chromatin remodelers can influence transcription factors that are already bound to DNA.

Here, Li et al. studied the interactions between a transcription factor and the chromatin remodelers in yeast. The experiment used a piece of DNA that contained a bound transcription factor and a single nucleosome. Li et al. used a technique called 'single molecule DNA unzipping', which enabled them to precisely locate the position of the nucleosome and transcription factor before and after the nucleosome was remodeled. The experiments found that a chromatin remodeler called ISW1a moved the nucleosome away from the transcription factor, while a SWI/SNF chromatin remodeler moved the nucleosome towards it.

Significantly, Li et al. also found that a transcription factor is a major barrier to ISW1a's remodeling activity, suggesting that ISW1a may use transcription factors as reference points to position nucleosomes. In contrast, SWI/SNF was able to slide a nucleosome past the transcription factor, which led to the transcription factor falling off the DNA. Therefore, SWI/SNF is able to move transcription factors out of the way to deactivate genes.

Li et al. propose a new model for how chromatin remodelers can move nucleosomes and regulate transcription factors to alter gene activity. A future challenge will be to observe these types of activities in living cells.

influence the chromatin landscape. We hypothesize that nucleosome remodeling, without remodeler recruitment, may regulate the state of a bound TF. Specifically, a remodeler may attempt to move a nucleosome to or through a site pre-occupied by a TF. During such an encounter, the TF may be displaced, or it may act as a roadblock for nucleosome remodeling. Thus, chromatin remodeling may serve as an alternative mechanism to regulate transcription through its influence on a bound TF, and a bound TF may dictate the location of a remodeled nucleosome.

Here, we studied the influence of nucleosome remodeling on a bound TF in a single molecule assay. We used a DNA unzipping technique (*Jiang et al., 2005*; *Shundrovsky et al., 2006*; *Hall et al., 2009*; *Jin et al., 2010*; *Dechassa et al., 2011*; *Li and Wang, 2012*; *Inman et al., 2014*) to characterize the locations of a bound TF and a nucleosome simultaneously, on long DNA templates to near base pair accuracy. By examining the remodeling behavior upon encountering a bound TF, we determined that the relationship between TFs and nucleosome remodeling not only plays a critical role in nucleosome positioning, but also reveals a novel mechanism for how a TF can be dynamically recycled by nucleosome remodeling.

## Results

### Precise determination of the position of a transcription factor and a nucleosome

In this work, we needed to precisely locate the positions of a nucleosome and a TF before and after nucleosome remodeling. We thus employed the DNA unzipping technique (*Figure 1—figure supplement 1*), which has been demonstrated to be a powerful single molecule technique for accurate and precise determination of positions and strengths of DNA–protein interactions

(*Jiang et al., 2005*; *Shundrovsky et al., 2006*; *Hall et al., 2009*; *Jin et al., 2010*; *Dechassa et al., 2011*; *Li and Wang, 2012*; *Inman et al., 2014*). To evaluate the precision of this approach, we constructed a DNA template containing a single Gal4 sequence for binding to the Gal4 DNA-binding domain (Gal4DBD) and a single 601 nucleosome positioning sequence (601NPE) for uniquely positioning a nucleosome. Gal4DBD contains only the 147 amino acids of the N terminal domain of the Gal4 protein and does not have any known remodeler recruitment function.

*Figure 1* shows representative traces from unzipping DNA molecules without nucleosome remodeling. The top trace of *Figure 1* shows the result when the DNA template was unzipped starting from the Gal4 side. Both Gal4DBD (a single smaller peak) and a nucleosome (two clusters of larger peaks) were readily detected above the baseline of the corresponding naked DNA. The bottom trace shows the result when the DNA template was unzipped starting from the nucleosome side. Although the nucleosome unzipping signature was readily detectable, the unzipping signature of Gal4DBD was sometimes masked by that of the nucleosome. Therefore, it was often necessary to carry out unzipping experiments from both directions. Analysis of these unzipping signatures confirmed that unzipping mapped the position of the TF and the nucleosome to near base pair precision ('Materials and methods'; *Figure 1—figure supplement 2*; *Figure 1—figure supplement 3a*; *Figure 1—figure supplement 4*).

These unzipping experiments also revealed tight binding of Gal4DBD to its recognition sequence and slow dissociation. Under our experimental conditions, the equilibrium dissociation constant of Gal4DBD was determined to be 3.4 nM (*Figure 1—figure supplement 3b*). Our experiments were carried out with 95% of Gal4 sites bound to Gal4DBD. In addition, the bound Gal4DBD's lifetime was much longer than 1 hr (the typical duration of a single molecule experiment) (*Figure 1—figure supplement 3c*). For all experiments involving Gal4DBD, including those in *Figure 1*, Gal4DBD was allowed to equilibrate with the DNA, and remaining free Gal4DBD was then flushed from the sample chamber. Thus, subsequent remodeling reactions were carried out without free Gal4DBD in solution.

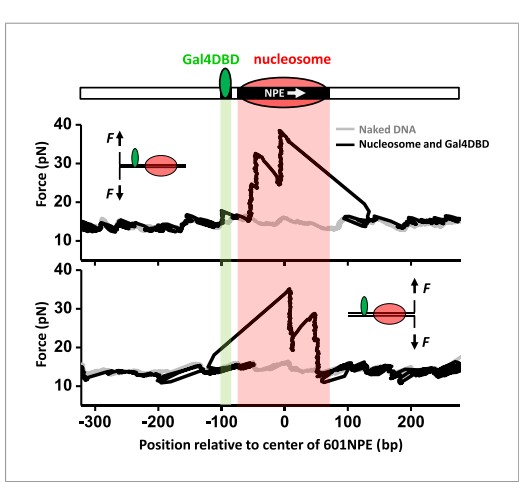

**Figure 1**. Single molecule unzipping technique detects Gal4DBD and nucleosome to near base-pair accuracy. DNA molecules, each containing a nucleosome and a bound Gal4DBD, were unzipped. All unzipped DNA molecules used in this work were in the region of 600 bp to 1.2 kbp. For clarity, much smaller regions are shown in all figures, with the origin of a template sequence defined as center position (the dyad) of the 601NPE. Shaded regions indicate locations of the Gal4 binding sequence and the 601NPE. (top panel) Cartoon illustrating the unzipping template design using for this experiment. A Gal4 sequence was separated from a 601NPE by 10 bp. The orientation of the 601NPE sequence is indicated by a white arrow. (middle panel) Unzipping in the direction in which the bound Gal4DBD was encountered first. (bottom panel) Unzipping in the direction in which the nucleosome was encountered first.

The following figure supplements are available for figure 1:

**Figure supplement 1**. Unzipping experimental configuration.

**Figure supplement 2**. Characterization of the precision and accuracy of detection of the locations of Gal4DBD and nucleosome.

**Figure supplement 3**. Characterization of Gal4DBD binding.

**Figure supplement 4**. Detection of Gal4DBD binding.

## Gal4DBD affects the directionality of ISW1a and SWI/SNF remodeling differently

Chromatin remodelers utilize ATP hydrolysis to move nucleosomes by altering histone–DNA interactions, with the two major families of chromatin remodelers, ISWI and SWI/SNF, differing in their outcomes of the remodeling reaction (*Clapier and Cairns, 2009*). SWI/SNF family remodelers are known to be associated with nucleosome disruption (*Imbalzano et al., 1996*; *Logie and Peterson, 1997*; *Aoyagi et al., 2002*) and transcriptional

activation (*Kwon et al., 1994*; *Hassan et al., 2001*; *Gkikopoulos et al., 2011*); while ISWI family remodelers have been shown to contribute to the formation of evenly spaced nucleosome arrays (*Tsukiyama et al., 1999*; *Fyodorov and Kadonaga, 2002*; *Lusser et al., 2005*; *Torigoe et al., 2011*). Despite these opposing characteristics, both remodeler families have been implicated in transcriptional activation and repression (*Kwon et al., 1994*; *Moreira and Holmberg, 1999*; *Hassan et al., 2001*; *Martens and Winston, 2002*; *Whitehouse et al., 2007*; *Yadon et al., 2010*). On mononucleosome substrates, many ISWI remodelers have been shown to be sensitive to naked DNA segments flanking the nucleosome, preferentially sliding the nucleosome towards the longer segment of DNA (*Yang et al., 2006*; *Blosser et al., 2009*; *Deindl et al., 2013*). This sensitivity to linker DNA is believed to underlie their ability to generate evenly spaced nucleosomal arrays (*Gelbart et al., 2001*; *Stockdale et al., 2006*). SWI/SNF remodelers, on the other hand, can shift a histone octamer up to 50 bp off the end of a short DNA fragment (*Kassabov et al., 2003*). On dinucleosomal templates, SWI/SNF remodelers have been found to shift one nucleosome onto another, indicating nucleosome disruption and eviction characteristics of these remodelers (*Engeholm et al., 2009*; *Dechassa et al., 2010*).

Here, we employed ySWI/SNF and yISW1a as model systems to study how a bound Gal4DBD may affect the remodeling of an adjacent nucleosome. First, we investigated the initial direction of nucleosome remodeling in the presence of a bound Gal4DBD in close proximity. This was achieved by limiting the remodeling reaction to the first remodeling event which we define as a single round of remodeler binding, nucleosome remodeling, and remodeler detachment from the nucleosomal DNA (*Shundrovsky et al., 2006*). We engineered a DNA template in which a Gal4 binding sequence and a 601NPE were separated by 10 bp. The DNA template containing a positioned nucleosome was then remodeled, by either SWI/SNF or ISW1a, for a short period of time (~1 min), with or without the addition of Gal4DBD (*Figure 2*; *Figure 2—figure supplement 1*; 'Materials and methods'). During such a short remodeling time, ~56% of nucleosomes pooled from measurements of multiple single molecules were found to remain at the original location, suggesting a lack of remodeling (*Figure 2—figure supplement 2*). Of the remaining ~45% of the nucleosomes that were remodeled, we estimate that ~73% were remodeled only once and ~27% were remodeled more than once, using a method we previously established (*Shundrovsky et al., 2006*).

In the absence of Gal4DBD, although the nucleosome unzipping signature did not appear to be altered after remodeling by either ISW1a or SWI/SNF (*Figure 2—figure supplement 1*; *Shundrovsky et al., 2006*), the positions of the nucleosomes were spread out from the original location. Both ISW1a and SWI/SNF were able to move a nucleosome bi-directionally (*Figure 2B,C*) without inducing significant changes in the nucleosome structure (*Figure 2—source data 1*). The slight asymmetric distribution of the remodeled nucleosome was likely due to the non-palindromic feature of the 601 sequence (*Lowary and Widom, 1998*) which leads to some asymmetry in the protein–DNA interactions at the two halves of a nucleosome (*Hall et al., 2009*). The results from SWI/SNF remodeling were also consistent with those from an earlier study (*Shundrovsky et al., 2006*).

Interestingly, in the presence of Gal4DBD, ISW1a moved the nucleosome away from Gal4DBD (*Figure 2D*), whereas SWI/SNF moved the nucleosome towards Gal4DBD (*Figure 2E*). To determine whether such a differential behavior was a result of the DNA sequence used, we engineered another DNA template that was identical to this one, except that the Gal4 binding site was located on the other side of the 601NPE. After adding Gal4DBD, ISW1a again moved the nucleosome away from the Gal4DBD (*Figure 2F*), while SWI/SNF again moved the nucleosome towards the Gal4DBD (*Figure 2G*). These data rule out the possibility of a DNA sequence effect on the directionality of nucleosome movement by the two remodelers. Therefore, we conclude that the bound Gal4DBD affects the directionality of nucleosome movement by the two types of remodelers differently: away from the TF for ISW1a and toward the TF for SWI/SNF.

Our findings on the TF directed SWI/SNF nucleosome remodeling are entirely novel; while our findings on the TF directed ISW1a nucleosome remodeling are in agreement with a previous study that used NURF (a homolog of ISWI complexes in Drosophila) in the presence of Gal4DBD (*Kang et al., 2002*).

## Gal4DBD is a barrier for nucleosome sliding by ISW1a

Since ISW1a moved a nucleosome away from an adjacent Gal4DBD, the bound Gal4DBD may provide a barrier to ISW1a remodeling. To test this, we designed an unzipping template with a 601NPE at the end of the template and a Gal4 binding at a greater spacing (75 bp) from the 601NPE

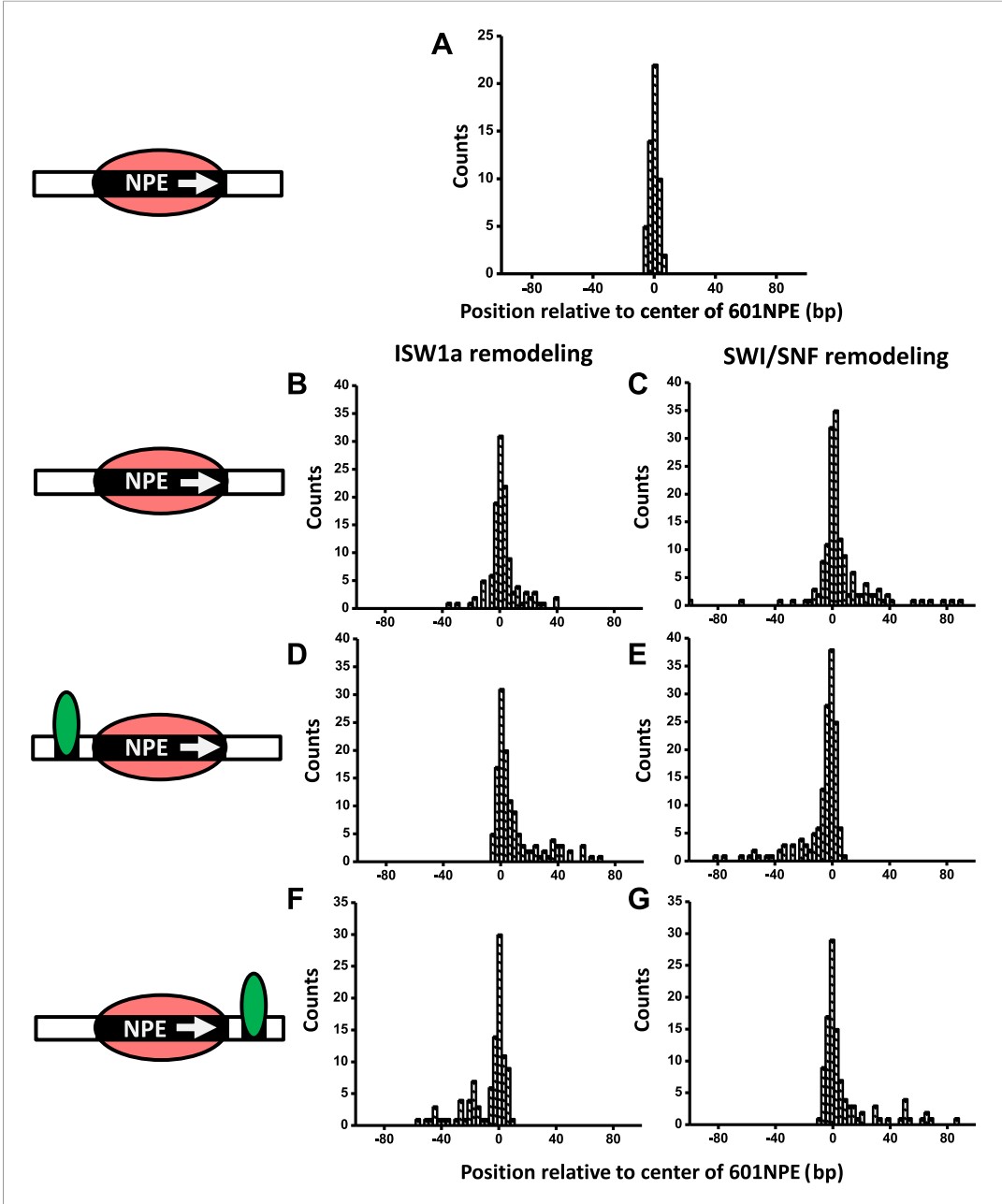

**Figure 2**. A bound Gal4DBD affects the directionality of SWI/SNF remodeling and ISW1a remodeling differently. Nucleosomes were remodeled by either 1 nM ISW1a or 1 nM SWI/SNF with 1 mM ATP for 1 min, a time sufficiently short that the majority of nucleosomes were not remodeled (*Figure 2—figure supplement 2*). Each DNA template was subsequently unzipped. For templates used in (**D**)–(**G**), the 601NPE was separated from the Gal4 binding sequence by 10 bp. (**A**) Distribution of the location of a nucleosome before remodeling. Data were pooled from measurements on multiple nucleosomal DNA molecules. (**B**) Distribution of the location of a nucleosome remodeled by ISW1a in the absence of Gal4DBD. (**C**) Distribution of the location of a nucleosome remodeled by SWI/SNF in the absence of Gal4DBD. (**D**) Distribution of the location of a nucleosome remodeled by ISW1a with a bound Gal4DBD initially located upstream of the 601NPE. (**E**) Distribution of the location of a nucleosome remodeled by SWI/SNF with a bound Gal4DBD initially located upstream of the 601NPE. (**F**) Distribution of the location of the nucleosome remodeled by ISW1a with a bound Gal4DBD initially located downstream of the 601NPE. (**G**) Distribution of the location of a nucleosome remodeled by SWI/SNF with a bound Gal4DBD initially located downstream of the 601NPE.

*Figure 2. continued on next page*

*Figure 2. Continued*

The following source data and figure supplements are available for figure 2:

**Source data 1**. Comparison of unzipping force signatures of a nucleosome before and after remodeling.

**Figure supplement 1**. Directionality of ISW1a and SWI/SNF remodeling of a nucleosome in close proximity to a bound Gal4DBD.

**Figure supplement 2**. Determination of fractions of remodeled nucleosome.

(*Figure 3—figure supplement 1*). The use of an end-positioned nucleosome should dictate that the nucleosome movement could only take place towards a bound Gal4DBD. After ISW1a remodeling for 10 min, which was sufficiently long to allow for multiple rounds of remodeling, the distributions of the nucleosome location showed a significant difference between the absence and presence of Gal4DBD (*Figure 3A,B*). In the absence of Gal4DBD, the nucleosome was moved away from the template end by several hundred base pairs towards the center region of the template, generating a rather broad distribution. In contrast, in the presence of Gal4DBD, although nucleosomes were still moved away from the end of the template, they were not able to pass the location of the Gal4DBD (*Figure 3B*). Instead, the distribution peaked at the midpoint between the Gal4 binding sequence and the 601NPE.

To further examine the relationship between the location of Gal4DBD and the ISW1a-remodeled nucleosome, we used two additional templates with shorter (24 bp and 50 bp) distances between the Gal4 binding site and the 601NPE (*Figure 3B*). After ISW1a remodeling in the presence of Gal4DBD, we found that the dyad locations of the remodeled nucleosomes always nearly centered between the bound Gal4DBD and the template end (*Figure 3C*; *Figure 3—figure supplement 2*). These results demonstrate that Gal4DBD is a physical barrier for ISW1a mediated nucleosome remodeling and that ISW1a is able to use Gal4DBD as a reference point to reposition a nucleosome.

This novel finding is of particular relevance to in vivo nucleosome spacing, especially near transcription start and termination sites. Although previous studies have suggested a possible role for ISWI remodelers to space nucleosomes using bound TFs near these sites (*Pazin et al., 1997*; *Gkikopoulos et al., 2011*; *Yen et al., 2012*), our results provide direct evidence that ISWI remodelers can indeed sense and respond to the presence of a DNA-bound protein such as a TF, which acts as barrier to dictate the placement of nucleosomes.

## SWI/SNF remodeling evicts Gal4DBD from DNA

In order to test whether a bound Gal4DBD is a physical barrier for SWI/SNF remodeling and what the fate of a bound Gal4DBD is upon nucleosome remodeling, we used the same template as the one that we used in single round experiments and performed 10 min remodeling on template both in the absence and in the presence of Gal4DBD. After a single round of nucleosome remodeling, the nucleosome will likely overlap the Gal4 binding site (*Figure 2*). However, the force signature of a bound Gal4DBD is subtle compared to that of a nucleosome (*Figure 1*), and thus the presence or the absence of a bound Gal4DBD cannot be definitively differentiated from a nucleosome by the unzipping force. Therefore, in order to determine whether a Gal4DBD was present after nucleosome remodeling, we allowed SWI/SNF to carry out multiple rounds of remodeling reactions to potentially reposition the nucleosome sufficiently far from the Gal4 binding site to allow for a definitive assay of the state of binding at the Gal4 binding site.

For this experiment, we designed a long template (∼1200 bp) with a 601NPE separated from a Gal4 binding sequence by 11 bp. The 601NPE was located near the center of a long DNA template to allow ample distance for possible bidirectional sliding of the nucleosome via multiple rounds of remodeling, such that the remodeled nucleosome and possible presence of Gal4DBD could be independently detected. After a 10 min remodeling by SWI/SNF, nucleosomes were repositioned from the center of the template to random locations along the entire sequence. Both in the absence and presence of Gal4DBD, remodeled nucleosomes were detected on both sides of the original Gal4 binding position (*Figure 4A*, *Figure 4—figure supplement 1*). This indicates that Gal4DBD is not a physical barrier for SWI/SNF mediated nucleosome remodeling.

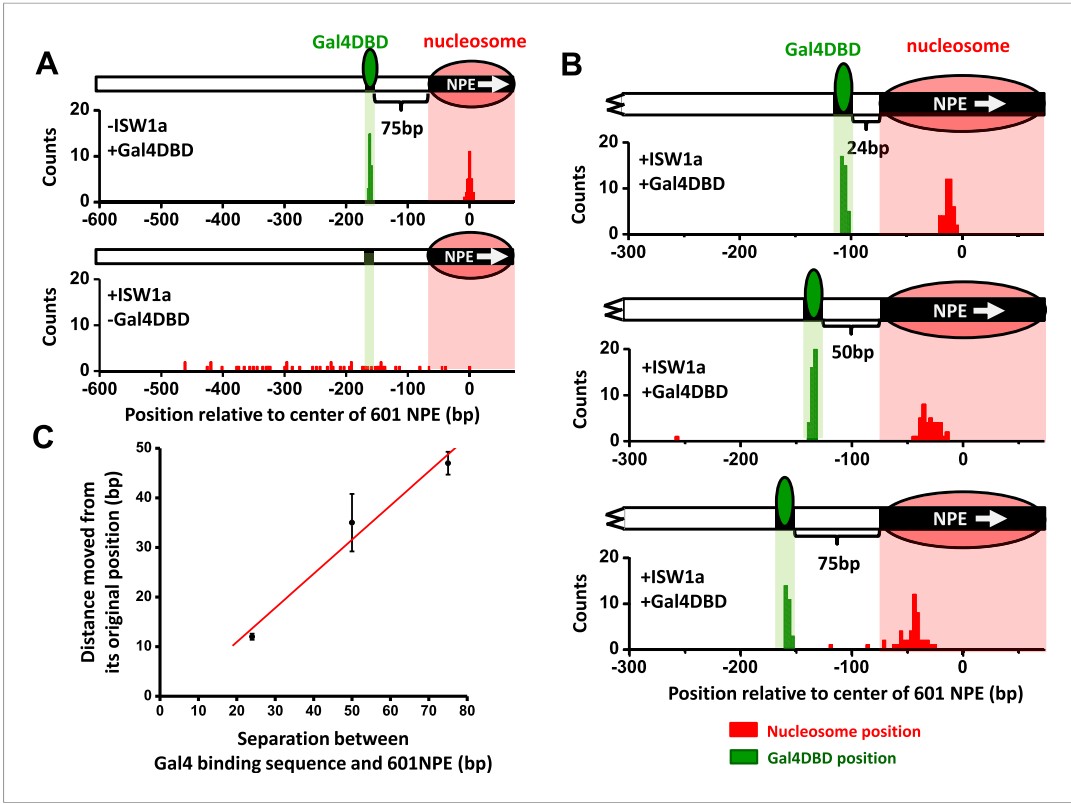

**Figure 3**. ISW1a remodeling is blocked by a bound Gal4DBD. Nucleosomes were remodeled by 1 nM ISW1a with 1 mM ATP for 10 min with or without Gal4DBD. Shaded regions indicate locations of Gal4 binding sequence and 601NPE. (**A**) Distributions of the locations of the nucleosome and bound Gal4DBD, either before remodeling or without Gal4DBD, as controls. (**B**) Distributions of the locations of the nucleosome after ISW1a remodeling in the presence of Gal4DBD on three different templates of increasing separation between the Gal4 binding sequence and the 601NPE. For each template, the nucleosome position distribution is dominated by a narrow population, but has a few outliers which were moved a much greater distance and some of which even passed the Gal4 binding sequence. These outliers (~5%) were likely a result of templates that did not have a bound Gal4DBD initially (~5%; see main text). This is further supported by the observation that none of these outlier traces revealed a bound Gal4DBD. Nonetheless, in order to avoid possible bias, these nucleosome positions were still used for further analysis in (**C**) and thus contributed to the error bars in (**C**). (**C**) Relationship between the distance the remodeled nucleosome moved and the separation between the Gal4 binding sequence and the 601NPE. Error bars are SEM.

The following figure supplements are available for figure 3:

**Figure supplement 1**. Single molecule unzipping simultaneously detects Gal4DBD and an end-positioned nucleosome.

**Figure supplement 2**. ISW1a centers a nucleosome between a bound Gal4DBD and the template end.

What is the fate of the Gal4DBD after a nucleosome has been remodeled? To answer this question, we allowed a nucleosome to be remodeled by SWI/SNF in the presence of Gal4DBD. We then analyzed each trace to determine whether a nucleosome was remodeled to the opposite side of the Gal4 sequence or to the same side of the Gal4 sequence, relative to the 601NPE. For the traces where nucleosomes were remodeled to the opposite side of the Gal4 sequence, we did not detect any Gal4DBD unzipping signature (*Figure 4B*; *Figure 1—figure supplement 4*). This indicates that SWI/SNF was able to move the nucleosome in such a way that the Gal4DBD was evicted from its binding sequence. For traces where a nucleosome was remodeled to the same side of the Gal4 sequence relative to the 601NPE (*Figure 4B*), we also did not detect any Gal4DBD unzipping signature. This implies that these nucleosomes were

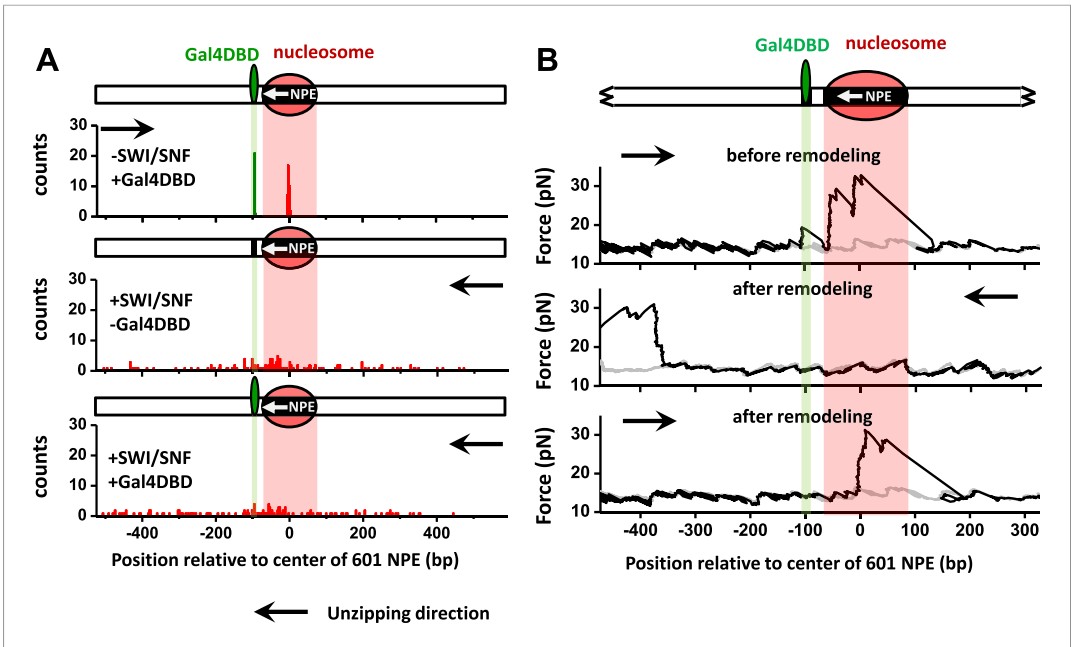

**Figure 4**. SWI/SNF remodeling evicts a bound Gal4DBD from its DNA template. Nucleosomes were remodeled by 1.5 nM SWI/SNF with 1 mM ATP for 10 min with or without Gal4DBD. Shaded regions indicate locations of Gal4 binding sequence and 601NPE. (**A**) Distributions of the locations of the nucleosome and bound Gal4DBD before remodeling (upper plot), after remodeling without Gal4DBD (middle plot), and after remodeling with Gal4DBD (lower plot). (**B**) Representative traces in the case of before remodeling (top plot; $N = 55$) and after remodeling (middle and bottom plots; $N = 50$). The middle plot shows an example trace where a nucleosome was remodeled to the opposite side of Gal4DBD relative to its original position; while the bottom plot shows an example trace where a nucleosome was remodeled to the same side of Gal4DBD relative to its original position. Gray traces were taken from the corresponding naked DNA.

The following source data and figure supplements are available for figure 4:

**Source data 1**. SWI/SNF is unable to evict a bound Gal4DBD in the absence of a nucleosome or in the absence of ATP.

**Figure supplement 1**. Distributions of the locations of SWI/SNF remodeled nucleosomes as determined by unzipping from both directions.

**Figure supplement 2**. Nucleosome remodeling by SWI/SNF on a template with the Gal4 binding site separated from the 601NPE by 24 bp.

likely first remodeled towards the bound Gal4DBD, as indicated by *Figure 2*, resulting in the eviction of Gal4DBD. This was followed by subsequent remodeling events that moved nucleosomes to other positions. Control experiments SWI/SNF is unable to evict a bound Gal4DBD in the absence of a nucleosome or in the absence of ATP (*Figure 4—source data 1*). An increase in the separation between the Gal4 binding site and 601NPS to 24 bp still permits SWI/SNF remodeling to evict Gal4DBD (*Figure 4—figure supplement 2*). Therefore, we conclude that SWI/SNF remodeling is able to evict a bound TF and this eviction requires the presence of a nucleosome.

Our findings provide the first direct evidence that SWI/SNF nucleosome remodeling is capable of evicting a bound TF. Previously, SWI/SNF was shown to move one nucleosome to invade and eventually disrupt an adjacent nucleosome (*Dechassa et al., 2010*). Taken together with our findings, SWI/SNF nucleosome remodeling appears to be powerful machinery capable of actively overcoming and removing a variety of obstacles in its vicinity.

## Nucleosome remodeling shows similar effects in the presence of a Lac repressor

To rule out the possibility that the interaction between a TF and a nucleosome demonstrated above is specific to Gal4DBD, we replaced Gal4 binding sequence with a Lac repressor binding sequence and repeated the above experiments using the Lac repressor. Because the Lac repressor is only found in prokaryotic cells and has no known relationship with any chromatin remodeler in eukaryotic cells, it can act as a biologically neutral bound protein. When ISW1a remodeled an end-positioned nucleosome on a template also containing a bound Lac repressor, the Lac repressor effectively dictated the position of the remodeled nucleosome, with ISW1a centering the nucleosome on the DNA with the Lac repressor acting as a barrier (*Figure 5A*). In contrast, SWI/SNF was able to slide a nucleosome in either direction, displacing the bound Lac repressor (*Figure 5B*). Therefore, we conclude that the mechanism of TF regulation by nucleosome remodeling is likely general without any specificity to a particular TF.

## Discussion

It has been widely acknowledged that SWI/SNF remodelers move nucleosomes toward the ends of a DNA template and ISWI remodelers move nucleosomes away from the ends of a DNA template (*Stockdale et al., 2006*; *Yang et al., 2006*; *Zofall et al., 2006*; *Leonard and Narlikar, 2015*). However, rather than DNA ends, in vivo DNA nucleosome remodeling will most likely encounter DNA-binding proteins such as TFs. Here, we show that ISW1a senses a bound TF as a boundary during nucleosome repositioning, while SWI/SNF remodeling is unimpeded by the presence of a bound TF and is able to slide a nucleosome and evict the TF (*Figure 6*). In contrast to bulk biochemical studies

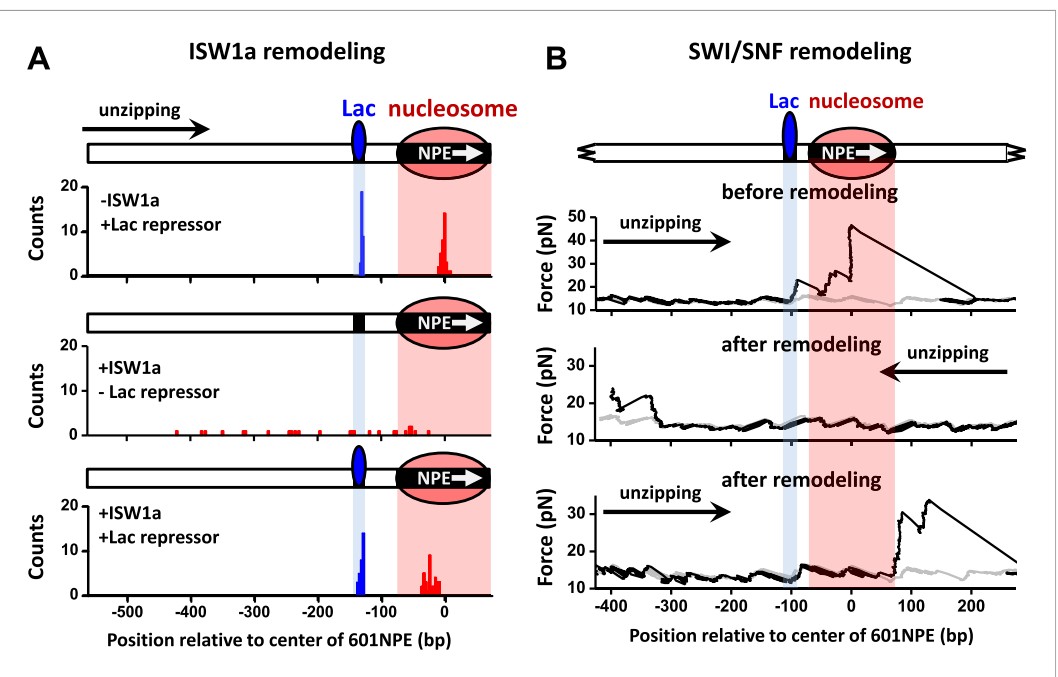

**Figure 5**. ISW1a remodeling is blocked by Lac repressor, while SWI/SNF remodeling evicts Lac repressor from the template. Shaded regions indicate locations of Lac repressor binding sequence and 601NPE. (**A**) Distributions of locations of nucleosomes before remodeling (upper plot), after remodeling by ISW1a without Lac repressor (middle plot), and after remodeling by ISW1a with Lac repressor (lower plot). Lac repressor binding sequence was separated from the 601NPE by 50 bp. Nucleosomes remodeling was carried out in 1 nM ISW1a with 1 mM ATP for 10 min with or without Lac repressor. (**B**) Representative traces in the case of before SWI/SNF remodeling (top plot; $N = 25$) and after remodeling (middle and bottom plots; $N = 27$). Lac repressor binding site was separated from the 601NPE by 10 bp. Nucleosomes were remodeled by 1.5 nM SWI/SNF with 1 mM ATP for 10 min. The middle plot shows an example trace where a nucleosome was remodeled to the other side of the Lac repressor and the bottom plot shows an example trace where a nucleosome was remodeled to the same side of Lac repressor.

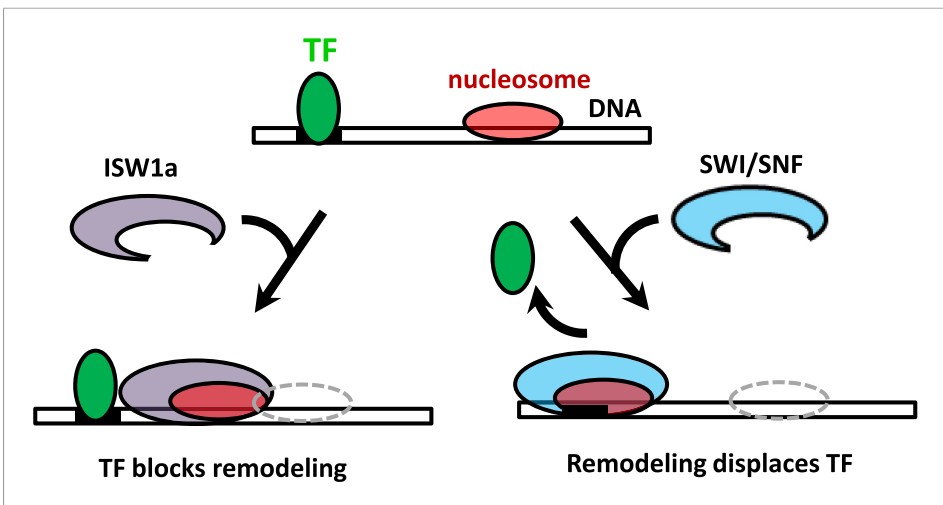

**Figure 6**. The interplay between nucleosome remodeling and a bound TF. When a nucleosome is remodeled by ISW1a (left), nucleosome repositioning is blocked by a bound TF, the TF remains intact, and the remodeled nucleosome is positioned with the TF acting as a boundary and reference point. On the other hand, when a nucleosome is remodeled by SWI/SNF (right), nucleosome positioning is unaffected by a bound TF and the TF is evicted.

that typically use DNA ends as boundaries on short DNA templates, our studies using bound TFs as potential barriers better mimic a situation that will more likely occur in vivo.

## Differential directionality of nucleosome remodeling

Our results showed opposite directionality for nucleosome positioning when a nucleosome in close proximity to a TF was remodeled by ISW1a and SWI/SNF (*Figure 2*). Biochemical studies and the crystal structure of ISW1a indicate that the DNA-binding domain of ISW1a binds to ~29 bp of the extranucleosomal DNA, which has been proposed to act as an anchor to pull the nucleosome towards ISW1a (*Gangaraju and Bartholomew, 2007*; *Hauk and Bowman, 2011*; *Yamada et al., 2011*; *Hota et al., 2013*). Our finding that ISW1a moves nucleosome away from a TF adjacent to the nucleosome is consistent with an important role of the DNA-binding domain in engaging fully accessible DNA immediately flanking the nucleosome. In contrast, while it has been widely acknowledged that SWI/SNF does not require extra-nucleosomal DNA binding to remodel a nucleosome, cryo-EM and DNA-crosslinking experiments have shown that the Snf6 subunit of SWI/SNF binds to ~15 bp of the extra-nucleosomal DNA and the rest of the SWI/SNF slides the nucleosome away from where the Snf6 subunit binds (*Dechassa et al., 2008*). Although the Snf6 subunit has not been shown to be essential for remodeling, it has DNA-binding affinity (*Sengupta et al., 2001*; *Dechassa et al., 2008*). We speculate that it may help to orient SWI/SNF binding on the nucleosome. In the presence of a barrier adjacent to a nucleosome, Snf6 may prefer to bind to the side of the nucleosome with more linker DNA and thus orient SWI/SNF to slide a nucleosome towards the TF.

## Nucleosome spacing by ISW1a in vivo

Our study of ISW1a remodeling demonstrates that Gal4DBD is an effective barrier for ISW1a-mediated nucleosome remodeling and the ISW1a is able to use Gal4DBD as a reference point to reposition nucleosomes (*Figure 3*). These results have significant implications for the mechanism of nucleosome spacing in vivo. Genome-wide nucleosome mapping in budding yeast revealed that deletion of ISWI in yeast disrupts nucleosome spacing (*Gkikopoulos et al., 2011*), suggesting that ISW1 remodelers are key players in generating evenly distributed nucleosomal arrays. In addition, several recent studies have shown that certain DNA-binding factors located at the promoter region are also responsible for nucleosome positioning (*Whitehouse et al., 2007*; *Yadon et al., 2010*; *Zhang et al., 2010*; *Bai et al., 2011*; *Hughes et al., 2012*; *Yen et al., 2012*; *Parikh and Kim, 2013*;

*Struhl and Segal, 2013*; *Lieleg et al., 2014*). Our finding that a bound Gal4DBD is a barrier to ISW1a now provides direct evidence to illustrate that ISW1a can potentially use a TF around the promoter region as a reference point to evenly position nucleosomes into the gene body.

### In vivo implications of SWI/SNF action in TF regulation

Our study shows that TF eviction is an intrinsic property of SWI/SNF remodeling and is independent of SWI/SNF recruitment (*Figures 4 and 5*). It has been previously shown that SWI/SNF recruitment by the glucocorticoid receptor (GR) induced histone loss in nucleosomes and this was immediately followed by GR and SWI/SNF eviction from the template (*Nagaich et al., 2004*). Our current work demonstrates that, in the absence of remodeler recruitment, TF eviction via nucleosome remodeling can take place without substantial nucleosome loss. Although SWI/SNF can translocate along naked DNA (*Lia et al., 2006*; *Zhang et al., 2006*; *Sirinakis et al., 2011*), raising the possibility for TF eviction solely by SWI/SNF, we found that in the absence of a nucleosome, SWI/SNF did not displace Gal4DBD from its binding site (*Figure 4—source data 1*). Thus, SWI/SNF translocation alone is insufficient to displace a bound Gal4DBD and TF eviction requires nucleosome remodeling. Previous work by Owen-Hughes and coworkers (*Lia et al., 2006*) found that translocation by RSC was highly sensitive to a force in the DNA. Therefore, although SWI/SNF is known to translocate along naked DNA, it may have limited ability in dealing with a road block, such as a bound protein. It is also possible that SWI/SNF is unable to efficiently locate a bound protein in the absence of a nucleosome. We speculate that TF removal may be accelerated once a nucleosome is repositioned over the bound TF. Indeed, a recent single molecule fluorescence study of Gal4 binding kinetics on nucleosomal DNA showed that a nucleosome regulates Gal4 binding not only by preventing Gal4 binding, but also by dramatically increasing the Gal 4 dissociation rate from the DNA (*Luo et al., 2014*).

SWI/SNF family remodelers are known to be involved in transcriptional activation. Genome-wide mapping of yeast indicates that, apart from localizing to nucleosomes around transcription start sites, SWI/SNF family remodelers are also enriched upstream of the promoter regions (*Yen et al., 2012*). Genome-wide analysis of the locations of human chromatin remodelers also found that Brg1, Chd4, and Snf2h are highly enriched at the promoter and distal upstream regions (*Morris et al., 2014*). Since many relevant transcriptional modulators, such as enhancers (*Ren, 2010*) and insulators (*Bell et al., 2001*), are located further upstream of promoters, SWI/SNF family remodelers could move promoter nucleosomes to dynamically regulate these factors. Thus, although SWI/SNF alone does not possess any ability to remove TFs on its own, our work shows that SWI/SNF can slide nucleosomes to displace neighboring TFs around the promoter region, providing a mechanistic basis for dynamically clearing both nucleosomes and other bound factors upon SWI/SNF recruitment (*Nagaich et al., 2004*).

## Materials and methods

### Plasmids

The plasmids containing the Gal4 binding site and the 601NPE with varied distances were prepared using standard PCR and cloning methods. The cloning segments were generated by standard PCR from the 601 plasmid (*Lowary and Widom, 1998*) using special primers, one of which contains one Gal4 binding site. The distance between the primer containing the Gal4 binding site and the 601NPE determines the distance between the Gal4 binding site and 601NPE. Then, the PCR product was cloned into the pDrive vector (Qiagen, Valencia, CA). The finished constructs were confirmed by DNA sequencing.

### Nucleosome unzipping template

Nucleosomal DNA templates were prepared using methods similar to those previously described (*Koch et al., 2002*; *Li and Wang, 2012*). Briefly, each DNA construct consisted of two separate segments. A ~1.1 kbp anchoring segment was amplified, by PCR, from plasmid pRL574 using a digoxigenin-labeled primer and then subsequently digested with *Bst*XI (NEB, Ipswich, MA) to produce an overhang. The unzipping templates were amplified, by PCR, from the plasmids described above and amplified with a biotin-labeled primer, digested with *Bst*XI, and dephosphorylated using CIP (NEB, Ipswich, MA) to introduce a nick into the final DNA template. Nucleosomes were assembled from purified HeLa histones onto the unzipping fragment by a well-established salt dialysis method

(*Lee and Narlikar, 2001*). The two segments were joined by ligation immediately prior to use. This produced a complete template labeled with a single dig tag on one end and a biotin tag located 7 bp after the nick in one DNA strand.

## Nucleosome remodeling reaction

yISW1a and ySWI/SNF were purified as previously described (*Gangaraju and Bartholomew, 2007*; *Dechassa et al., 2008*). yGal4DBD was purchased from Santa Cruz Biotechnology, Inc., Dallas, TX. After the ligation of the anchoring segment and unzipping segment containing a nucleosome, we incubated 20 nM of the nucleosomal DNA with 200 nM Gal4DBD at 16°C for 30 min. Single molecule sample preparation was performed according to protocols previously described (*Li and Wang, 2012*). The remodeling experiments were conducted in a sample chamber after the DNA tethers are formed. SWI/SNF remodeling reactions contained 1.5 nM purified ySWI/SNF, and 1 mM ATP in the SWI/SNF remodeling buffer (10 mM Tris·Cl, pH 8.0, 100 mM NaCl, 7 mM MgCl$_2$, 2 mM DTT, 0.1 mg/ml acBSA). ISW1a remodeling reactions contained 1.5 nM purified yISW1a, and 1 mM ATP in the ISW1a remodeling buffer (30 mM HEPES, pH 7.6, 3 mM MgCl$_2$, 5 mM NaCl, 0.1 mM EGTA, 0.02 mM EDTA, 5% glycerol, 0.2 mg/ml acBSA). Both types of remodeling reactions were incubated at 25°C with duration specified. The reactions were stopped by the addition of 10 mM EDTA and 0.25 mg/ml Salmon Sperm DNA and incubation for 5 min at 25°C. Finally, the sample chamber was rinsed with 100 µl sample buffer (10 mM Tris·Cl pH 7.5, 1 mM EDTA, 100 mM NaCl, 1.5 mM MgCl$_2$, 1 mM DTT, 3% (vol/vol) glycerol, 0.02% (vol/vol) Tween 20, and 2 mg/ml BSA). Single molecule unzipping measurements were subsequently performed in this sample buffer.

## Data collection and alignment

An optical trapping setup as previously described (*Brower-Toland and Wang, 2004*) was used to unzip a single DNA molecule by moving the microscope coverslip horizontally away from an optical trap. The unzipping methods have been previously described (*Li and Wang, 2012*) and briefly summarized here. Whenever the unzipping fork encountered an interaction that prevented the fork progression, the unzipping force was ramped up linearly with time (15 pN/s) until the interaction was disrupted. When two interactions occurred in close vicinity, upon the disruption of the first interaction the force was unable to relax back to the baseline before being ramped up again for the second interaction, subjecting this subsequent interaction to a higher initial force. Therefore, for each region of interactions, the dwell time histogram highlighted the edge of the region first encountered. Another feature of this method was the display of the distinctive force signature for a nucleosome, allowing for robust identification of the nucleosome structure.

Data were low pass filtered to 5 kHz, digitized at ~12 kHz, and later low pass filtered to 60 Hz. The precision and accuracy of the experimental curves were improved to near base pair level by cross-correlation of regions immediately before the Gal4DBD disruption and after the nucleosome disruption, using methods as previously described (*Hall et al., 2009*; *Li and Wang, 2012*). For the experimental curves where the nucleosomes are located at the end of the template, the cross-correlation was carried out for a region immediately before the Gal4DBD disruption or nucleosome disruption. To account for minor instrumental drift, trapping bead size variations, and DNA linker variations, the alignment allowed for a small additive shift (~10 bp) and multiplicative linear stretch (<2%) using algorithms similar to those previously described (*Hall et al., 2009*).

## Determination of locations of a TF and a nucleosome

Gal4DBD showed a distinct unzipping signature with a single force peak at 8 bp from the center of the consensus sequence (*Figure 1—figure supplement 2* and *Figure 1—figure supplement 3a*), indicating the front end of the Gal4DBD footprint on the DNA. The disruption force peak was 18–20 pN, significantly larger than the baseline force of ~15 pN. Therefore, we determined the center position of a bound Gal4DBD by first detecting the peak force location and then shifting this location by 8 bp in the direction of unzipping.

The positioned nucleosome displayed a much more complex force signature with multiple force peaks and a significantly greater overall force, reflecting the multiple finer and stronger histone–DNA interactions within a nucleosome (*Shundrovsky et al., 2006*; *Hall et al., 2009*). We determine the dyad position of a nucleosome by first measuring mean force location within the first force cluster and then shifting this position by 43 bp in the direction of the unzipping (*Figure 1—figure supplement 2*).

## Acknowledgements

We thank members of the Wang lab for critical reading of the manuscript. We especially thank Dr RM Fulbright for purification of the histones, and Dr JT Inman, Dr SM Fellman, Dr RA Forties, and Ms LD Brennan for technical advice with data acquisition and plasmid cloning. We wish to acknowledge support to MDW from the National Institutes of Health grant (GM059849) and National Science Foundation grant (MCB-0820293) and to GDB from National Institutes of Health grant (R01-GM084192).

## Additional information

### Funding

| Funder | Grant reference | Author |
| --- | --- | --- |
| National Institute of General Medical Sciences (NIGMS) | GM059849 | Michelle D Wang |
| National Science Foundation (NSF) | MCB-0820293 | Michelle D Wang |
| National Institute of General Medical Sciences (NIGMS) | GM084192 | Gregory D Bowman |

The funders had no role in study design, data collection and interpretation, or the decision to submit the work for publication.

### Author contributions

ML, Conception and design, Acquisition of data, Analysis and interpretation of data, Drafting or revising the article; AH, PS, Purified and characterized yeast SWI/SNF; LO, Purified and characterized yeast ISW1a; MAH, Upgraded the optical trapping setup and wrote software for data conversion and analysis; BYS, Upgraded the optical trapping setup and wrote software for data conversion; SF, Upgraded the optical trapping setup; JNMK, Participated in the initial experiments that laid the foundation for the current work; AP, Purified and characterized yeast the lac repressor; GDB, BB, Conception and design, Drafting or revising the article; MDW, Conception and design, Analysis and interpretation of data, Drafting or revising the article, Contributed unpublished essential data or reagents

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
