## [Decision Letter]

Thank you for sending your work entitled “Dynamic Regulation of Transcription Factors by Nucleosome Remodeling” for consideration at *eLife*. Your article has been overall favorably evaluated by James Manley (Senior editor), two expert reviewers, and a member of our Board of Reviewing Editors.

The Reviewing editor and the two reviewers discussed their comments before we reached this decision, and the Reviewing editor has assembled the following comments to help you prepare a revised submission.

In this manuscript, Li et al. use a single-molecule DNA unzipping approach to probe the mechanism of nucleosome remodeling by two ATP-dependent nucleosome-remodeling complexes, Isw1a and SWI/SNF, and the influence of DNA-binding protiens on this process. While the biochemical properties and mechanisms of action of ATP-dependent remodeling factors have been studied extensively, relatively few studies have examined how DNA-binding proteins impact and are impacted by nucleosome remodeling. Using a previously described unzipping assay, the authors present evidence that a Gal4 derivative (1-147) or the Lac repressor can block nucleosome sliding by ISWI1a. By contrast, the data suggest that SWI/SNF action leads to eviction of both Gal4 and LacR. The authors find that DNA-binding proteins serve as a barrier to nucleosome sliding by Isw1a, and nucleosomes tend to become evenly spaced between the DNA-binding protein and a barrier on the other side (such as the end of the DNA fragment). By contrast, SWI/SNF preferentially remodels nucleosomes toward the DNA-binding protein and displaces the DNA-binding factor in the process. This displacement by SWI/SNF depends on nucleosome remodeling. The experimental results as shown appear solid, but there are several concerns with the experimental design that create serious issues as described below.

Specific major comments:

1) Figure 1: Given the modest peak corresponding to Gal4DBD relative to naked DNA, some statistical analysis of the peak height distribution across multiple traces with and without Gal4DBD binding is necessary to understand how confidently the Gal4DBD-bound vs unbound state can be assessed in later experiments.

2) In Figure 4, the authors allow SWI/SNF to remodel nucleosomes for 10 minutes, observing that Gal4DBD is removed from the template in most cases, including those in which the nucleosome ends up at a location further away from the Gal4 binding site. Since the authors already observed that nucleosomes are initially moved by SWI/SNF closer to the Gal4DBD, they infer that nucleosomes are initially moved near or past the Gal4 binding site, displacing Gal4DBD, and then move in either direction afterward. This seems like a missed opportunity—why not just examine what happens to Gal4DBD in the initial remodeling event (in a 1 minute, or slightly longer, remodeling reaction as performed in Figure 2)? This would allow assessment of whether the nucleosome actually needs to invade the Gal4 binding site to displace the DBD, or whether loss of DBD occurs prior to that, when the nucleosome gets within a certain distance of the DBD. This additional mechanistic insight would add some novelty and strengthen the paper.

3) It is not clear how SWI/SNF complex can translocate past the Gal4DBD without displacing it in the absence of a nucleosome (in the first paragraph of the subsection headed “In vivo implications of SWI/SNF action in TF regulation”). Do the authors think the Gal4DBD is temporarily displaced when SWI/SNF moves past and then re-binds, that the DBD remains bound when SWI/SNF translocates past its binding site, or that SWI/SNF cannot translocate past a bound DBD in the absence of nucleosomes? Some discussion of this point would be useful.

4) Gal4 (1-147) is known to contain a transcriptional activation domain that is functional in vitro (Lin et al., Cell. 1988 Aug 26;54(5):659-64). This derivative is never used as a DNA binding domain control, but rather the 1-94 derivative is employed. SWI/SNF is known to be recruited by acidic activation domains, and it is not clear if targeting/binding by Gal4 (1-147) is impacting these results, especially the directionality of remodeling. If the authors wish to further investigate the directionality issue, they should either use 1-94, or use the Lac repressor constructs.

5) The authors need to provide an ATP control to demonstrate that loss of the Gal4 signature reflects actual remodeling events, rather than simply a consequence of SWI/SNF binding. This is especially problematic as the SWI/SNF experiments all appear to use substrates where the Gal4 or Lac binding sites are extremely close to the nucleosome edge (10-11 bp). In contrast, the ISW1a experiments employ a variety of templates with varying distances. It is not clear why the same templates were not used for SWI/SNF studies. From their model, SWI/SNF should be able to evict Gal4 even if it is located at great distances from the 601 sequence. Such templates should be tested.

6) Note that previous nucleosome mapping experiments by the Bartholomew lab (11) showed that the Snf6 subunit of SWI/SNF actually binds to the first 15 bp of linker DNA and would thus compete for Gal4 (1-147) or Lac binding.

7) Note that older experiments from the Kadonaga group ([51], Science 276:809) showed very nicely that a DNA binding domain (tetR) could provide a boundary function for ATP-dependent remodeling enzymes. Although these older studies used a *Drosophila* embyro extract system, we now know that these assays were monitoring the activity of the ISWI-containing ACF complex. These older studies detract from the novelty of this current study.

8) The authors state that the ability of Gal4 to impart directionality onto the SWI/SNF remodeling reaction is novel. However, Workman and colleagues have previously provided evidence for a similar effect in ensemble experiments with Gal4-VP16 (EMBO J 2007 26:730).

9) It is not clear from the Methods section when the unzipping reaction is performed. Are remodeling enzymes stopped and removed before the unzip or after? Please explain.

10) Please explain why were constructs with factor binding sites directly adjacent to the nucleosomes the only ones used in the SWI/SNF studies. Were these constructs the only ones that worked?

---

## [Author Response]

*1)*
Figure 1*: Given the modest peak corresponding to Gal4DBD relative to naked DNA, some statistical analysis of the peak height distribution across multiple traces with and without Gal4DBD binding is necessary to understand how confidently the Gal4DBD-bound vs unbound state can be assessed in later experiments*.

We thank the reviewers for this suggestion. The statistical analysis of the Gal4DBD was presented in the original manuscript as the old Figure 4—figure supplement 2 (Detection of Gal4DBD binding). As seen in the histograms of the detected force at the Gal4 binding site, Gal4DBD binding yielded 21 ± 0.8 pN (mean ± s.d.), whereas the corresponding naked DNA yielded 15 ± 0.5 pN (mean ± s.d.). This clear difference in the force magnitude allowed reliable differentiation of Gal4DBD binding from the naked DNA baseline.

Upon reading this comment, we realized that our initial placement of this supplement figure in the manuscript might not have been optimal. We have moved it forward in this modified manuscript so that it is now associated with Figure 1 and is referred to as Figure 1—figure supplement 4 (Detection of Gal4DBD binding).

*2) In*
Figure 4*, the authors allow SWI/SNF to remodel nucleosomes for 10 minutes, observing that Gal4DBD is removed from the template in most cases, including those in which the nucleosome ends up at a location further away from the Gal4 binding site. Since the authors already observed that nucleosomes are initially moved by SWI/SNF closer to the Gal4DBD, they infer that nucleosomes are initially moved near or past the Gal4 binding site, displacing Gal4DBD, and then move in either direction afterward. This seems like a missed opportunity*—*why not just examine what happens to Gal4DBD in the initial remodeling event (in a 1 minute, or slightly longer, remodeling reaction as performed in*
Figure 2*)? This would allow assessment of whether the nucleosome actually needs to invade the Gal4 binding site to displace the DBD, or whether loss of DBD occurs prior to that, when the nucleosome gets within a certain distance of the DBD. This additional mechanistic insight would add some novelty and strengthen the paper*.

We appreciate this thoughtful comment. We agree with the reviewers that it would be interesting to gain additional mechanistic insight to determine whether the loss of a bound Gal4DBD occurs either due to the invasion of the nucleosome or prior to this invasion. We had indeed contemplated the possibility of examining the fate of Gal4DBD during the initial remodeling event. Unfortunately we concluded that this experiment does not appear to be feasible for the following reason. As shown in Figure 2, after the initial remodeling event, SWI/SNF repositions the nucleosome by ∼ 30 bp towards a bound Gal4DBD so that the nucleosome will likely overlap the Gal4 binding site (Figure 2). However, the force signature of a bound Gal4DBD is subtle compared to that of a nucleosome (Figure 1), and thus the presence or the absence of a bound Gal4DBD cannot be definitively differentiated from the nucleosome by the unzipping force. Therefore, in order to determine whether a Gal4DBD was present after nucleosome remodeling, we allowed SWI/SNF to carry out multiple rounds of remodeling reactions to potentially reposition the nucleosome sufficiently far from the Gal4 binding site to allow for a definitive assay of the state of binding at the Gal4 binding site. We have now revised the main text to clarify this point (subsection headed “SWI/SNF remodeling evicts Gal4DBD from DNA”).

*3) It is not clear how SWI/SNF complex can translocate past the Gal4DBD without displacing it in the absence of a nucleosome (in the first paragraph of the subsection headed “In vivo implications of SWI/SNF action in TF regulation”). Do the authors think the Gal4DBD is temporarily displaced when SWI/SNF moves past and then re-binds, that the DBD remains bound when SWI/SNF translocates past its binding site, or that SWI/SNF cannot translocate past a bound DBD in the absence of nucleosomes? Some discussion of this point would be useful*.

We appreciate that the reviewers brought up this interesting point. We did not state in the manuscript that “SWI/SNF complex can translocate past the Gal4DBD without displacing it in the absence of a nucleosome”. However, the ability of SWI/SNF to translocate along DNA raises the possibility that SWI/SNF might be able to displace a TF on its own. Our control experiment was indeed designed to investigate this possibility and we found that SWI/SNF, in the absence of a nucleosome, did not displace a TF. We thus concluded that “SWI/SNF translocation alone is insufficient to displace a bound Gal4DBD, and TF eviction requires nucleosome remodeling” (please see “Differential directionality of nucleosome remodeling”).

Previous work by Owen-Hughes and coworkers (37) found that translocation by RSC was highly sensitive to a force in the DNA. Therefore, although SWI/SNF is known to translocate along naked DNA, it may have limited ability in dealing with a road block, such as a bound protein. It is also possible that SWI/SNF is unable to efficiently locate a bound protein in the absence of the nucleosome. We have added some discussion on this under the Discussion section (subsection headed “In vivo implications of SWI/SNF action in TF regulation”).

*4) Gal4 (1-147) is known to contain a transcriptional activation domain that is functional in vitro (Lin et al., Cell. 1988 Aug 26;54(5):659-64). This derivative is never used as a DNA binding domain control, but rather the 1-94 derivative is employed. SWI/SNF is known to be recruited by acidic activation domains, and it is not clear if targeting/binding by Gal4 (1-147) is impacting these results, especially the directionality of remodeling. If the authors wish to further investigate the directionality issue, they should either use 1-94, or use the Lac repressor constructs*.

We agree that Gal4 (1-94) is a more stringent DNA binding domain control than Gal4 (1-147). We used Gal4 (1-147) for the following reasons:

A) We had in fact considered using Gal4 (1-100) initially. However, Gal4 (1-100) was shown to have 50 times lower binding affinity to the Gal4 binding site than Gal4 (1-147): *K*d is 500 pM for Gal4 (1-100) (Liang, S. D. et al. MCB 1996) and 20 pM for Gal4 (1-147) (Carey et al. JMB 1989). Our single molecule experiments were carried out over a course of up to one hour. Since a lower DNA binding affinity is often correlated with a shorter dwell time on DNA, we were concerned about the possibility of a short dwell time of Gal4 (1-100). We found that Gal4 (1-147) is stably bound to DNA without any substantial dissociation for the entire period of experiments in the absence of nucleosome remodeling by SWI/SNF.

B) Gal4 (1-147) has been shown not to activate transcription in vivo (Field, S. and Song, O. Nature 1989), nor does it have any known capability to recruit nucleosome remodelers. In the publication by Lin et al. (Cell, 1988), experiments were not conducted on nucleosomal DNA templates and transcription activation appeared to be mediated by a mechanism that does not involve SWI/SNF recruitment.

C) Bartholomew and colleagues (9) found that Gal4-VP16 recruitment of SWI/SNF biases a nucleosome away from the bound Gal4-VP16. We observed the opposite effect: SWI/SNF moves a nucleosome towards the bound Gal4 (1-147), indicating a lack of specific recruitment of SWI/SNF by Gal4 (1-147).

D) Although we used Gal4 (1-147) as the main TF in our system, we also complemented this work with the Lac repressor, which is a foreign protein not present in the eukaryotic system. The fact that we observed essentially identical results using both Gal4 (1-147) and the Lac repressor further suggests that Gal4 (1-147) served primarily as a DNA binding domain.

*5) The authors need to provide an ATP control to demonstrate that loss of the Gal4 signature reflects actual remodeling events, rather than simply a consequence of SWI/SNF binding. This is especially problematic as the SWI/SNF experiments all appear to use substrates where the Gal4 or Lac binding sites are extremely close to the nucleosome edge (10-11 bp). In contrast, the ISW1a experiments employ a variety of templates with varying distances. It is not clear why the same templates were not used for SWI/SNF studies. From their model, SWI/SNF should be able to evict Gal4 even if it is located at great distances from the 601 sequence. Such templates should be tested*.

We appreciate that the reviewers raised the possibility that the loss of a bound Gal4DBD might be a result of SWI/SNF binding, rather than SWI/SNF remodeling of the nucleosome. We have thus conducted an additional experiment to examine this possibility by repeating experiments similar to those presented in the original Figure 4, except in the absence of ATP. As shown in the new [Supplementary-material SD2-data] (right column), in the absence of ATP, SWI/SNF does not result in Gal4DBD dissociation, even when the Gal4 binding site is separated by 10 bp from the 601NPE. This revised figure further demonstrates the Gal4DBD eviction requires the simultaneous presence of SWI/SNF, ATP, and nucleosome.

We also appreciate the question as to why our experiments with SWI/SNF were not conducted on a variety of templates with the Gal4 binding site separated at a range of distances from the 601NPE, as were done with ISWI experiments. This was due to technical challenges with the SWI/SNF experiments, which might not be evident from the original manuscript. In contrast to ISW1a-remodeled nucleosomes that were always located within a rather narrow distribution on one side of the TF, SWI/SNF remodeled nucleosomes were located over a much broader range of positions on both sides of the Gal4 binding site. For a given remodeling reaction time, the farther the Gal4 binding site is separated from the 601NPE, the lower the probability of detecting a remodeled nucleosome that is repositioned to the other side of the Gal4 binding site. In addition, in order to determine whether the Gal4DBD is still present on the template when a nucleosome is repositioned to the other side of the Gal4 binding site, the nucleosome must be located sufficiently far from the Gal4 binding site so that the footprint of the repositioned nucleosome does not overlap with the Gal4 binding site (see our answer to comment #2). Thus the larger the separation between the Gal4 binding site and the 601NPE, the more data traces that must be collected to accumulate sufficient statistics of such events.

That being said, we recognize the importance of providing further evidence that a bound TF, located further away from the 601NPE, can still be evicted by SWI/SNF nucleosome remodeling. Therefore, we have conducted an additional experiment on a template where the Gal4 binding site was separated from the 601NPE by 24 bp. The results from this new experiment confirm that if a nucleosome was repositioned to the other side of the Gal4 binding site, Gal4DBD was evicted. These results are now presented as the new Figure 4—figure supplement 2.

*6) Note that previous nucleosome mapping experiments by the Bartholomew lab (*[11]*) showed that the Snf6 subunit of SWI/SNF actually binds to the first 15 bp of linker DNA and would thus compete for Gal4 (1-147) or Lac binding*.

We thank the reviewers for noting the Snf6’s binding to DNA and its potential effect on our results. Because the Snf6 subunit binding requires ∼ 15 bp of linker DNA, the 10 bp separation between the Gal4 binding site and the 601NPE is likely insufficient for SWI/SNF binding. Thus we feel that this constraint may lead SWI/SNF to orient itself so that Snf6 will preferably bind to the other side of the nucleosome. According to the work from the Bartholomew lab (11), the ATPase domain binds on the opposite side of the Snf6 binding location on the nucleosome, and we believe that the preferential binding of Snf6 to the relatively open side of the nucleosome eventually leads the SWI/SNF remodeling towards the bound TF. Text commenting on this can be found in the subsection headed “Differential directionality of nucleosome remodeling”.

It is worth pointing out that this preferential orientation of SWI/SNF binding on a nucleosome is not required for TF eviction. As demonstrated in the new Figure 4—figure supplement 2 in response to comment #5, the 24 bp separation between the Gal4 binding site and the 601NPE should allow Snf6 subunit binding, and this configuration still results in TF eviction.

*7) Note that older experiments from the Kadonaga group (*[51]*, Science 276:809) showed very nicely that a DNA binding domain (tetR) could provide a boundary function for ATP-dependent remodeling enzymes. Although these older studies used a* Drosophila *embyro extract system, we now know that these assays were monitoring the activity of the ISWI-containing ACF complex. These older studies detract from the novelty of this current study*.

We agree that the work by the Kadonaga group ([51], Science 276:809) provided one of the earlier evidences that a DNA bound protein can affect the nucleosome landscape. However, the use of a *Drosophila* extract in that work made it difficult to determine which protein, or proteins, contributed to this function. Considering our findings, that SWI/SNF and ISWI family enzymes yield differential outcomes in dealing with a bound TF, we feel it is essential to study these two classes of enzymes separately. The novelty of our work lies at the findings of the highly different behaviors of these two families of enzyme, in terms of the direction to reposition a nucleosome adjacent to a bound TF, the positions of the remodeled nucleosomes, and the fate of the TF after remodeling. Nonetheless, we recognize the importance of this earlier work by the Kadonaga group, and it is now referenced in the subsection “Gal4DBD is a barrier for nucleosome sliding by ISW1a”.

*8) The authors state that the ability of Gal4 to impart directionality onto the SWI/SNF remodeling reaction is novel. However, Workman and colleagues have previously provided evidence for a similar effect in ensemble experiments with Gal4-VP16 (EMBO J 2007 26:730)*.

We appreciate this point regarding the novelty of our work and would like to emphasize the difference between our work and Workman’s:

A) The Workman and colleagues paper focused on octamer dissociation by SWI/SNF, stimulated by Gal4-VP16. In contrast, our work focused on the fate of Gal4DBD. We also did not detect substantial octamer dissociation.

B) In terms of remodeling directionality, our findings are very different from those of Workman and colleagues. In the Workman experiment, the Gal4 binding site, separated by ∼20 bp from the nucleosome, was used to recruit SWI/SNF via a bound Gal4-VP16. Under this configuration, nucleosomes were remodeled off the template, away from the Gal4 binding site. In contrast, our template design placed the Gal4 binding site 10 bp from the 601NPE, and we found that the nucleosome was remodeled towards the Gal4 binding site. We attribute the opposite directionality from these two studies to SWI/SNF orientation on the nucleosome. In the Workman experiment, the 20 bp separation was sufficient for the binding of Snf6 subunit of the SWI/SNF, and thus the Gal4-VP16 oriented the SWI/SNF to slide the nucleosome away from the Gal4 binding site. In our experiment, the 10 bp separation prevented binding of Snf6 subunit so that the Snf6 could only bind on the site of the nucleosome distal to the Gal4 binding site. This difference in directionality also highlights the difference in nucleosome remodeling with and without specific recruitment.

C) We believe that our work provides unique insight about how SWI/SNF can further regulate a bound protein after being recruited by a specific TF. Our work complements Workman’s and provides a more comprehensive picture of how transcription can be regulated by SWI/SNF nucleosome remodeling.

*9) It is not clear from the Methods section when the unzipping reaction is performed. Are remodeling enzymes stopped and removed before the unzip or after? Please explain*.

Unzipping experiments were conducted after a remodeling reaction was quenched and the remodelers were removed from the sample chamber. We have slightly revised the Methods section to clarify this. Please see the subsection headed “Nucleosome remodeling reaction” for detailed experimental conditions and procedures.

*10) Please explain why were constructs with factor binding sites directly adjacent to the nucleosomes the only ones used in the SWI/SNF studies*. *Were these constructs the only ones that worked?*

This question is related to comment #5 and we provide a detailed response there. In short, we have conducted a new experiment using a larger separation between the Gal4 binding site and the 601NPE and these new results are now shown as Figure 4—figure supplement 2.